# Assessment of Occupational Safety and Hygiene Perception among Afro-Caribbean Hair Salon Operators in Manchester, United Kingdom

**DOI:** 10.3390/ijerph16183284

**Published:** 2019-09-06

**Authors:** Haruna Musa Moda, Debrah King

**Affiliations:** Department of Health Professions, Manchester Metropolitan University, Manchester M15 6BG, UK

**Keywords:** Afro-Caribbean, hairstylist, pollution exposure, safety and hygiene awareness, ergonomic risk factors

## Abstract

Because of exposure to a number of potential health hazards within the work environment, hairstylists experience occupational diseases that include occupational asthma, skin conditions and musculoskeletal diseases. The paucity of studies assessing occupational safety and hygiene management among Afro-Caribbean hair salon operators in the UK promoted the study. Qualtrics^TM^ was used to assess the participants’ perception of exposure to hair products and their personal safety and hygiene knowledge, attitudes, awareness, and risk perceptions at work. In five salons, indoor air quality was monitored over one working week for selected environmental pollutants: temperature, humidity, CO, CO_2_ and Total Volatile Organic Compounds (TVOCs) using a GrayWolf Direct Sense Indoor Air Quality-IAQ (IQ-610). The use of unflued gas heating to raise the indoor temperature was common among the salons’ operators which explains the high carbon monoxide readings recorded. Itchy eyes and nose (44.4%) shoulder, neck and back pain (39.2%) were frequently reported. Age-stratified analysis of reported occupational ailments showed participants within an age bracket of 31–35 reported allergies (24%) and itchy eyes and nose (19.1%) as the most common of occupational ailments. Respiratory, skin and musculoskeletal symptoms ranked as major occupational ill-health experiences among the study population. The study outcome demonstrated that the type of activity and the hair products used play an important role in the level of pollutants in the working environment. The substitution of the more harmful hair products with safer alternatives is needed, as is the encouragement of health surveillance.

## 1. Introduction 

Haircare is not seen as a high-risk profession, however the job can lead to occupational diseases that include skin conditions, musculoskeletal diseases (arthritis and tendonitis) and work-related asthma as a result of occupational exposure to numerous potential health hazards within the work environment including vapours, solvents, perfumes and dust. Due to the nature of the business, salon operators are exposed to a cocktail of chemicals through both their skin and respiratory system from hair dyes, bleach, shampoos, hair conditioners, hair relaxers, permanent wave solutions, detergents, hair spray and perfumes [1,2,3,4]. This exposure can likewise affect the customers and others present within the indoor environment where these cosmetic products are being used [4]. An inventory of cosmetic ingredients reported by the European Union and other researchers revealed that products used in the industry contain over 5000 volatile substances that are either harmful, irritants or toxigenic [4,5,6,7]. Lysdal [8] added that hairdressers are generally exposed to many substances well known to cause undesirable effects to the respiratory system. Compounds such as persulfates in cosmetic products have been known to cause delayed asthma and instant skin reactions, such as allergic and irritant contact dermatitis. Earlier work by Bradshaw et al. [9] reported incidences of self-reported asthma as well as chest tightness and coughing amongst the hairdressers surveyed, with 86% of the reported incidences being among females with an average age of 27. 

Due to the nature of hair products, their frequent use exposes the operators to harmful chemicals which are either known or suspected allergens, carcinogens or organic solvents that arise from activities including hair dyeing, shampooing, hair conditioning, hair sprays, hair relaxing, hair bleaching, Brazilian blowout and many more [2,8]. Similarly, Sranton [10] commented on the rise in the use of hair straighteners containing formaldehyde. A study by Monakhova et al. [11] examined 10 different hair treatment products and found seven of these products contained formaldehyde at levels higher than the recommended safe amount. In addition, evidence has established a link between exposure of volatiles released from hair treatment products with increased risks of reproductive disorders and low birth weight among babies as a result from the mother’s long-term exposure to these pollutants [11,12].

Several studies suggest that hairdresser salon operators are more likely to experience skin disease conditions that include occupational dermatitis than people in other professions [12,13,14,15], as a result of occupational exposure to chemicals or prolonged wet work but the dermatitis may be triggered by other factors such as bacterial infections and chemical burns [16]. An earlier study by Bradshaw et al. [9] in the UK involving 60 hair-salon operators, found a third of the participants have had hand dermatitis. A study by Tapper [17] reported 70% of hairdressers surveyed in Britain had suffered from work-related dermatitis, mainly to the hands and fingers but also to the arms, face and neck.

Another health concern affecting hairdressers is musculoskeletal discomfort, pain or injury, which can reduce job performance and productivity, lead to increased time off work and even early retirement [18,19]. Early work conducted in the UK [20], reported that significant numbers of hairdressers experience upper limb soft tissue disorders including carpal tunnel syndrome, tendonitis and back problems. Further to this, hair salon operators are more likely to come in contact with blood through processes such as hair carving, shaving cutting, manicure, pedicure, and skincare and where good hygiene management is lacking such acts can serve as a vehicle in the spread of transmittable disease among clients [21,22,23].

The aim of the research was to assess occupational exposure to indoor pollutants and the level of occupational safety and hygiene awareness amongst Afro-Caribbean salon operators. 

## 2. Importance of the Study Rationale

According to the European Agency for Safety and Health at Work, there are around 400,000 registered hair salon companies in Europe and the industry attracts about 350 million potential clients yearly [24]. While there is research that has examined the occupational exposure and safety hazards within the hair salon industry in general [25,26,27], to the best of our knowledge there has been limited study of occupational hygiene and safety management among Afro-Caribbean hair salon operators. Compared with Caucasian hair, African heritage hair is stronger and thicker. The Afro-Caribbean salon operators are at a higher risk of exposure to pollution than their Caucasian counterparts because of the nature and amount of materials used in treating their customers’ hair. The majority of hair treatments have been proven to contain chemicals considered harmful to humans. In addition, much longer hours are spent on styling the clients’ hair than would be the case with Caucasian clients, leading to long hours of standing and maintaining postures that could result in long term impacts on the stylist. 

## 3. Materials and Methods

### 3.1. Selection of Participants

Two methods were adopted in identifying participants to take part in the questionnaire survey. The first approach was through in-person recruitment of salon operators and the second using social media i.e., WhatsApp, Facebook and email, where a link to the survey was shared for each participant to complete the survey online. To ensure voluntary participation, a consent form with information on the study was made available to each participant giving an overview of the purpose of the project, what was expected from them in the project, their right to refuse or withdraw at any time, how their data would be used and managed, and their relevant legal rights. The exclusion criteria adopted was that only Afro-Caribbean hair salon operators were selected for the survey. To achieve this, a consent of ethnicity was asked on the online form to help in the identification of target population. A further control measure was a question asking each participant to confirm they had operated a hair salon (barber and/or hair salon) for more than one year and had a history of exposure to volatile chemicals released from hair products. Only when these conditions had been agreed could participants access the questionnaire. Ethical approval for the project was given in accordance with the guidance of the host institution; Manchester Metropolitan University. 

### 3.2. Questionnaire Design 

The survey was a cross-sectional study conducted from May 2017 to February 2018 where standardised questionnaires were adapted from the European Agency for Safety and Health at Work, E- Facts 34 and modified to suit the study aim. Participants were asked questions about subjects that included occupational hygiene, safety knowledge, attitudes, awareness, and risk perceptions. Work-related symptoms in this context means ill-health that get worse at work or improves on rest days and the duration of symptoms stratified as current (within the last week) or within the last 3 months, as described by Bradshaw et al. [9]. 

The survey was divided into several sections that included: demographic characteristics, safety awareness at work, use of personal protective equipment (PPE) and chemical exposure risk, skincare and protection, prevalence of symptoms, ergonomic risk, and prevention management to reduce occupational hazards. 

### 3.3. Environmental Monitoring of Selected Variables 

Five hair salons, located within a 2-mile radius in Manchester and identified as suitable during the questionnaire survey, agreed to progress to this stage. They were randomly allocated dates for measurement of their indoor air quality during the months of March and April 2018. Three of the salons catered for men’s hair and the remaining three provided a service for women. At each salon, one-week of indoor air quality monitoring was carried out to monitor the distribution pattern and build-up of selected indoor air pollutants. The size of the salons ranged between 10–15 m^2^ with an average of two operators in each salon. 

Prior to the sampling of the selected pollutants, a visual inspection of each salon was conducted to establish the condition of the existing working environment. Equipment, materials, building layout and types of services rendered were audited as well as the types of hair treatment products displayed in each salon as these were considered likely to impact on the air quality in each salon. 

### 3.4. Environmental Sampling Methods 

Selected indoor air quality (IAQ) parameters (temperature, relative humidity, carbon dioxide, total volatile organic compounds and carbon monoxide) were monitored during the working hours over a Monday to Sunday period.

Sampling was performed using a GrayWolf Direct Sense IAQ (IQ-610) equipped with a multi-gas photo-ionization detector (PID) in accordance with software prescribed standard operating procedure. The PID was equipped with a part per billion (ppb) sensor with a measurement range of 5–20,000 (ppb) for Total Volatile Organic Compounds (TVOCs) 0–10,000 (ppm) for CO_2_, and 0–500 ppm for carbon monoxide. The IQ-610 displays measurements in real-time with data value recorded at 15-minute time interval for each measured parameter held on a pocket personal computer connected to the air-sampling probe. GrayWolf advance sense software version WS2013.11 was used to process the data generated. To guard against sensitivity loss from either sensor poisons or suppressors present in the indoor environment monitored, the instrument was factory calibrated before the study and then a functional (bump) test was conducted prior to the start of each sampling at each location to ensure that the response of the sensors lay within the acceptable tolerance range of the actual concentration. Prior to the start of each sampling, the equipment was programmed and allowed to warm up for 20 minutes to allow all sensors to stabilise and ensure accurate reading of each airborne pollutant in the salons. 

The data logger was placed in the carrier box at a height of one metre above the floor and away from any heating source in each salon during sampling. This allowed the sampling head to be exposed to the indoor air whilst protecting it from tampering. Normal day-to-day activity in each salon was encouraged while the data capture was ongoing. At the end of each monitoring cycle, data logged onto the portable logger was downloaded for analysis. Time-weighted average (TWA), short-term exposure limit (STEL), maximum- and minimum-logged values were worked out for each of the salons. General information was collected from the salon operators that related to the building use, activities carried out, use of hair treatment products and safety precautions. Both online survey and measured environmental data were analysed using Statistical Package for Social Sciences (SPSS) vs. 24 (IBM, New York, USA) and significance level set at 0.05.

## 4. Results

### 4.1. Questionnaire Analysis 

The survey was accessed by 153 respondents and 128 (83.7%) completed the survey. Among the participants that responded to the survey, 35 (22.9%) were male and 112 (73.2%) female, while the remaining six (3.9%) participants did not disclose their gender and 43.8% reported they have been working in the industry in the last five years (Table 1). In general, participants identified different forms of hair treatment including perming, dyeing, blow out, haircut, and hair-styling alongside manicure and pedicure as the most common activities performed on a day-to-day basis. Air extraction was not commonly used, with only 28 (18.3%) of the respondents stating that they had some form of extraction fan installed in their premises. 

From the results, complaints of itchy eyes and nose (44.4%) and back and neck pain (39.2%) were the most common forms of occupational disease or injury reported by the study group (Table 2). Long hours of standing while at work was agreed to be a major contributing factor to ankles and legs swelling. All participants stated that they had never used any form of respiratory protective equipment (RPE) even when they were performing tasks that involve the use of substances that release volatiles such as when perming, bleaching, shampooing or blow-drying. When questioned about their awareness of the health impacts posed by frequent inhalation of volatiles released from bleaching or perming products, 43 (28.1%) agreed that it could have long-term health impacts, however, the use of RPE had never been considered by any of the respondents. Contact dermatitis was reported by 47 (30.7%) of the participants. According to the respondents, hand gloves were not frequently used during wet activities because the multi-tasking nature of the job required putting on and taking off, thus 76 (49.7%) responded that they ‘do not fancy using them’. Some 37 (24.25%) of the participants stated that the frequent use of gloves made their hands sweat and that glove-use could cause allergic reactions. Accordingly, they were unhappy about using gloves despite manufacturers’ instructions to use gloves when handling their products. 

Based on age-stratified analysis of reported occupational ailments among participants in the study, those within the age bracket of 31–35 reported allergies (24%) followed by itchy eyes and nose (19.1%) as their most common forms of occupational illness. Contact dermatitis (23.4%), swollen ankles (28.6%) and joint ache (33.3%) were the most prevalent complaints registered by participants within the age range of 36–40 (Table 3).

### 4.2. Indoor Air Sampling

Five salon operators volunteered to have, their hair salon indoor air quality measured for Total Volatile Organic Compound (TVOC), CO_2_, CO, Temperature and Humidity. The salons had an average indoor space of about 33 m^3^ and physical assessment of each premise revealed that none had an extractor fan. In each salon, there was an average of two staff working during the physical assessment with the exception of salon two where there were six staff working on different activities, which included manicure, perming, barbing and nail polishing. 

Due to the time of the year (March–April), when the indoor air was measured all salons had their heating on and in a few instances, unflued gas heaters were used to regulate the indoor temperature which explained the high carbon monoxide reading in some of the salons (Table 4). Considering that the instrument used was calibrated with only one compound (isobutylene), the TVOC signal represents all compounds of the mixture as an equivalent of this compound. The output signal gives no information about the qualitative composition of the mixture. Closer examination of TVOC distribution over the monitoring duration showed varied distribution throughout the days. In all monitored salons, higher TVOC readings were at their peak close to the end of the working week (Figure 1). Salon 2 had the highest amount of TVOC released into the indoor environment with an average reading of 8215.9 ppb and 5316 ppb short-term exposure limit (STEL) and a time-weighted average (TWA) reading of 6955 ppb. Comparing the distribution pattern of the measured environmental variables for each sampling day, it was observed that, pollution build-up and peak levels differed in each salon, dependent on the number of clients and the type of work which had been undertaken for that day. Results from salon 1 showed that it had its highest TVOC peak of 2351 ppb recorded on Thursday. Temperature assessment revealed maximum temperatures in the salons ranged between 19.4 and 28.2 °C, whilst the relative humidity ranged between 48.4% and 59.2% respectively. The highest CO_2_ reading of 2463.8 ppm was recorded in salon 5. In the same salon, CO_2_ (STEL) was found to be 2239 ppm whilst the TWA during the same period was 1214 ppm (Table 4). The CO_2_ daily distribution pattern was associated with a number of factors including the number of staff and clients and the activity type/duration which had been undertaken.

The long-term exposure limit (8 hours) for temperature in most of the salons was found to be outside the recommendations contained in the Workplace (Health, Safety and Welfare) Regulations 1992: Approved Code of Practice L24 [28]. An indoor temperature of 16 °C should be considered as a minimum especially in a workplace where tasks undertaken do not involve severe physical effort. There is a need for the temperature inside the salon to provide reasonable comfort without the need for special clothing or where any such conditions could present discomfort to both workers and their clients. 

### 4.3. Impact of Work Environment on Pollutants Generation

High levels of TVOCs were generated as a result of the diverse range of activities undertaken including pedicures, manicures, acrylic nail fixing and hair perming. An online check of available safety data sheets (SDS) showed that manufacturers of some of the afro hair treatment products recommended their usage in well-ventilated areas. Where airborne concentrations are suspected to be above the permissible exposure limits use of approved respirators are recommended. These safety precautions and others recommended in the SDS were not operational in any of the salons visited. From physical assessment of salon 2 on the day of the instrument installation and decommissioning revealed the types and amount of hair products in use is a contributor to the much higher TVOC (8215.9 ppb) recorded in that salon (Table 4). To determine the role played by non-professional treatment factors considered likely to influence TVOCs concentration during 8 h TWA in each premises, a comparison was made with temperature and CO_2_ concentrations. It was found that elevated temperature was significantly and positively correlated with TVOC concentration in each salon (spearman R = 0.624; *p* < 0.01). High CO_2_ concentrations were found to be associated with elevated TVOC concentrations however, the association was not robust (R = 0.391, *p* < 0.01).

## 5. Discussion 

A number of occupational health problems associated with hair salon operators were considered in the study and the results highlighted the significance of these problems. There is a growing population of users of Afro-Caribbean salons in the UK and the paucity of up-to-date knowledge among the study participants is cause for concern. Respiratory, skin and musculoskeletal symptoms ranked as the predominant occupational ill-health concerns experienced by the study population. The study outcome has demonstrated that the types of activities and the composition of hair products used play an important role in the level of pollutants found in the indoor working environment. The level of pollution recorded in each working environment reflected the poorly ventilated indoor environment. This was partly due to the salon design which did not take the end-use into consideration so pollutants generated from different tasks tended to accumulate inside the premise. 

Over 43% of the online survey respondents reported having worked within the industry for under five years. There is a correlation between occupational and environmental hygiene awareness among the participants and the (low) use of respiratory protection equipment. The unawareness of the danger to health posed by the volatiles associated with the different tasks undertaken by the participants raises questions about the participants’ level of health and safety awareness when using products. Because of the season when the work was conducted, various forms of heating devices were in use. This included unflued gas heaters, which explained the high amount of pollutants, and temperature readings measured in each salon premise. Recommendations by the Chartered Institution of Building Services Engineers(CIBSE) stated that, as part of the minimum requirements for ventilation for hygiene and air quality in an open plan shop/salon, a fresh air supply rate of 8 litres per second per person should be maintained [29]. In addition, the Approved Code of Practice and Regulation L24 [28] required, in a situation where fixed heating systems are operational, the building operator should maintain the system so that the products of combustion do not enter the workplace and should have a sufficient air supply to ensure complete combustion so as not to produce fumes which will be harmful or offensive. Based on the aforementioned, there is the need for further study to assess the impact caused by the widespread use of unflued gas systems by salon operators and the indoor comfort and wellbeing of the operators and users of these salons. 

Earlier studies have demonstrated a direct relationship between insufficient fresh air in workplaces and health-related impacts that included tiredness, lethargy, headaches, dry or itchy skin and eye irritation among employees which was associated with the build-up of pollutants. These forms of ill health are aggravated due to the work being undertaken in poorly designed premises where unsatisfactory working conditions exist and the employees do not have control of certain aspects of their work [30,31]. These studies explained why respiratory and eyes irritation was prevalent among the present study participants, 44.4% of whom reported experiencing these symptoms while at work. Other effects may include sensory irritation, dryness and weak inflammatory irritation in the nose, airways and skin. In addition, Mølhave et al. [32] reported that where TVOC concentrations in the indoor environments exceed 25 mg/m^3^ this will present an increased chance of the occupants experiencing sensory effects, which may become of greater concern. 

Another occupational hygiene issue identified as prevalent among the participants was contact dermatitis. During the installation and decommissioning of sampling equipment, observations of the salon operators’ activity revealed a continuous wetting and drying of their hands during hairdressing procedures. This was in addition to the frequent application of cosmetic products used to treat the client’s hair. Previous studies have confirmed that some of the chemicals contained in these products are harmful and cause skin irritation and work-related dermatitis to large numbers of hair salon operators at some point [33,34,35,36]. In addition, the Health and Safety Executive (HSE) [35] report on work-related skin disease in Great Britain (2017) ranked hairdressers and barbers as having the second-highest rate of dermatitis by occupation with 72.5 cases per 100,000 workers per year between 2006–2016. Some 49.7% of the respondents said they did not use gloves during the application of hair products or for other tasks. If this is not addressed, the incidence of skin contact dermatitis will continue to rise amongst salon operators. The key reason for this observed trend was the failure to access the safety training and education which is provided by the UK Hair Council. Not one of the salon operators had undertaken such training. 

The different postures and movements required by different activities in a salon have been associated with a risk of joint injury among hairdressers in general. The present study showed that 27.5% of the respondents had experienced pain in the joints of the wrist, ankles or legs at some point and they associated these problems with the long hours of maintaining single postures and awkward movements of the joints. Relative high force exertion and wrist velocity along with prolonged time of exposure has earlier been identified in previous study by Chen [19] as major factors responsible for the high rate of hand/wrist pain experienced amongst salon operators. Furthermore, work-related problems, such as soft tissue conditions affecting the shoulder, elbow, forearm, wrist, thumb, hand and fingers alongside traumatic, degenerative, and inflammatory conditions (mostly of the legs and lower back) was found to be common among the cohort considered in previous reported studies [9,20]. Because salon workers operate with their arms raised at a 60° angle or more for approximately 13% of the total working time and 16% during the specific hairdressing tasks [18], this arm posture presented an increased risk of exacerbating the impact of neck and shoulder stresses on these operators, leading eventually to musculoskeletal injury. Whilst it may be difficult to conclude, based on the present study findings, that the reported musculoskeletal injuries were directly related to the tasks undertaken at work, there are grounds to associate the work undertaken with the reoccurring incidence of injury experienced amongst the cohort studied. 

## 6. Conclusions

This study adds to the existing body of literature that emphasizes the urgency of bringing to the attention of hair salon operators the need to act on the guidance for the reduction of exposure to harmful products and the ways of limiting the ergonomic impact of working within the industry in general. The findings showed that afro hair treatment products in combination with the type of work being carried out influenced the quantity of pollutants present in the work environment. This, in turn, affects the health of both the workers and the salon clients. There is a need for hair salon operators to consider changing their working habits as a means of improving indoor air quality. 

It is a known fact that inhalation of volatile chemical products present in the indoor work environment can present acute and chronic respiratory conditions among exposed staff. There is strong evidence associating occupational asthma and Chronic Obstructive Pulmonary Disease (COPD) with exposure to occupational and environmental chemicals. In this study, a large number of the respondents reported experiencing respiratory allergic reaction, dizziness and headache. 

The results of the monitoring of pollutants in the salons revealed that the high number of volatiles released to the indoor work environment explains the high incidence of reported occupational ailments such as allergic reactions, headache, dizziness, itchy eyes and nose. In addition, the frequent use of strong bleaching products, shampoos and hair spray on the clients contributes to the build-up of these pollutants and the resultant ill health reported. Findings from this study suggest that there is a lack of available data about the Afro-Caribbean salon operations and there is a need for a wider study to examine in depth the issues raised and promote the development of interventions to address work-related injury and sickness in the industry. 

### Recommendations

Considering the high percentage of workers that indicated they suffer from back and joint pain it is obvious that long periods of standing in the same position, maintaining undesirable postures and the movements involved with activities such as hair cutting, combing, washing, and blow-drying are major causes of musculoskeletal discomfort, pain or injury among hairdressers and barbers. There is a need to minimise this risk in order to reduce time taken off work or the injuries leading to early retirement from the trade. As part of health surveillance, it is important that workers are examined regularly to assess if exposure to work with arm elevation, prolong standing and maintaining undesirable postures are risk factors for back and shoulder injury. 

More effort is required to enlighten salon operators as to the dangers of prolonged exposure to airborne pollutants at work. Hence, the need for these operators to substitute the more harmful hair products with safer alternatives. Installation of extract fans to improve air quality should be considered and where possible mobile local exhaust ventilation could be considered to help extract pollutants and improve the indoor air quality.

## Figures and Tables

**Figure 1 ijerph-16-03284-f001:**
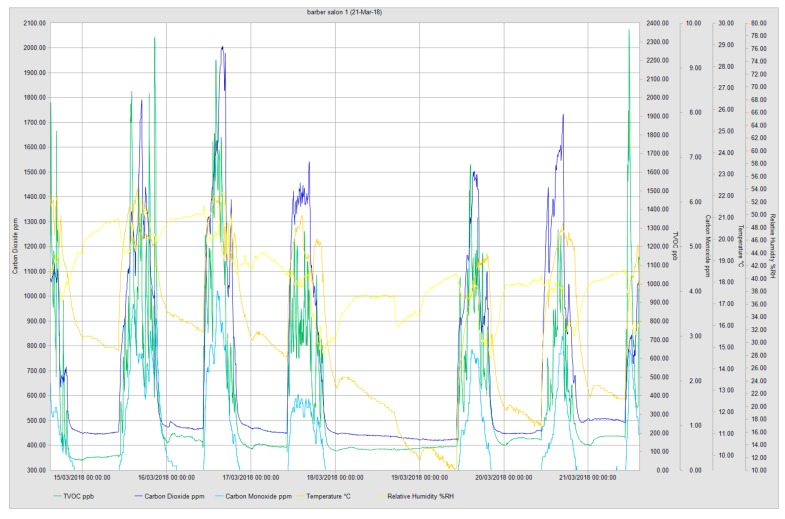
Example of real-time environmental monitoring in one of the salons showing the distribution pattern of monitored variables.

**Table 1 ijerph-16-03284-t001:** Demographic characteristics for operators of hair salons.

Characteristics	*n* (%)	Mean	Median	Standard Deviation
Gender				
Male	35 (22.9)	1.81	2	0.483
Female	112 (73.2)			
Prefer not to say	6 (3.9)			
Age in years				
18–20	12 (7.8)			
21–25	19 (12.4)	4.43	4	1.691
26–30	27 (17.6)			
31–35	30 (19.6)			
36–40	25 (16.3)			
41–45	17 (11.1)			
46–50	14 (9.2)			
5–above	6 (3.9)			
Prefer not to say	3 (2.0)			
Years working in the hair salon industry				
1–5	67 (43.8)			
6–10	49 (32)	2	2	1.187
11–15	17 (11.1)			
16–20	10 (6.5)			
21–above	10 (6.5)			

**Table 2 ijerph-16-03284-t002:** Occupational diseases suffered.

Occupational Disease/Injury	*n* (%)	Frequency of Occurrence (*n*)		
		Always	Sometimes	Most Time
Respiratory allergy	38 (24.8)	3	20	15
Contact dermatitis,	47 (30.7)	4	18	25
Itchy eyes and nose	68 (44.4)	5	36	27
Shoulder, neck and back pain	60 (39.2)	2	35	23
Joint aches	30 (19.6)	2	12	16
Headache, dizziness and nausea	57 (37.3)	4	27	26
Swollen joints: wrist, ankles and legs	42 (27.5)	5	19	18

**Table 3 ijerph-16-03284-t003:** Occupational ailments prevalence reported based on age distribution among the study population.

Age	Reported Occupation Aliments Reported (%)						
	Allergies	Itchy Eyes and Nose	Contact Dermatitis	Swollen Ankle and Leg	Joint Ache	Shoulder, Neck and Back Pain	Headache, Dizziness and Nausea
21–25	3 (7.9)	9 (13.2)	3 (6.4)	5 (11.9)	2 (6.7)	9 (15)	5 (8.8)
26–30	8 (21.1)	16 (24)	11 (23.4)	4 (9.5)	5 (16.7)	10 (16.7)	14 (24.6)
31–35	9 (24)	13 (19.1)	7 (14.9)	8 (19.1)	8 (26.7)	13 21.7)	12 (21.1)
36–40	8 (21.1)	12 (17.6)	11 (23.4)	12 (28.6)	10 (33.3)	13 (21.7)	12 (21.1)
41–45	7 (18.4)	9 (13.2)	7 (14.9)	8 (19.1)	3 (10)	9 (15)	5 (8.8)
46–50	3 (7.9)	3 (4.4)	4 (8.5)	2 (4.8)	1 (3.3)	4 (6.7)	5 (8.8)
51+	-	6 (8.8)	4 (8.5)	3 (7.1)	1 (3.3)	2 (3.3)	4 (7)
Total	38	68	47	42	30	60	57

**Table 4 ijerph-16-03284-t004:** Indoor environmental sampling.

Measured Compounds	Salon 1	Salon 2	Salon 3	Salon 4	Salon 5
TVOC (ppb)					
Min	52	2501	304	1182	168
Max	2351	15752	5909	8809	2428
Average	365.6	8215.9	1521.2	3330.9	2463.8
STEL (15 min)	1096	5316	2119	5851	291
TWA (8 h)	550	6955	907	2248	232
CO_2_ (ppm)					
Min	420	554	457	491	517
Max	2009	2124	2668	3006	7257
Average	695.1	1117.5	794.2	1117.1	2463.8
STEL (15 min) ***	1055	586	1783	1614	2239
TWA (8 h) ***	654	670	809	638	1214
CO (ppm)					
Min	0.0	0.3	0.0	0.7	0.2
Max	4.0	1.7	5.8	14.6	23.4
Average	0.57	0.94	1.08	4.42	6.8
STEL (15 min) **	0.8	0.3	1.6	5.4	7.5
TWA (8 h) **	0.8	0.6	0.4	2.0	3.4
Temperature (°C)					
Min	9.3	14.2	13.4	16.6	16.7
Max	22.3	19.4	28.2	24.6	24.8
Average	15.61	17.92	18.92	20.25	21.1
STEL (15 min)	17.8	17.2	24.3	21.8	22.8
TWA (8 h)	15.2	17.5	18.0	18.6	21.4
Relative Humidity (%RH)					
Min	27.4	39.8	25.3	29.1	43.0
Max	53.9	59.2	49.7	48.4	51.6
Average	40.42	41.58	34.11	36.54	45.57
STEL (15 min)	32.9	40.5	38.8	47.7	46.7
TWA (8 h)	37.8	40.9	39.5	42.9	46.7

Note: STEL: Short-term exposure limit. TWA: Time-weighted average. *** EH40 CO_2_ Long-term exposure limit (8-hour) TWA reference period = 5000 ppm and Short-term exposure limit (15-minute reference period) 15,000 ppm. ** CO Long-term exposure limit (8-hour) TWA reference period = 30 ppm and Short-term exposure limit (15-minute reference period) 200 ppm.

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
