# Peer review of "Assessment of Occupational Safety and Hygiene Perception among Afro-Caribbean Hair Salon Operators in Manchester, United Kingdom"

_ijerph, 2019, doi:10.3390/ijerph16183284_

Round 1
Reviewer 1 Report
1. My very first comment is on the type of manuscript: This manuscript is submitted to be published as a case report, while a case report is a "detailed report of the symptoms, signs, diagnosis, treatment, and follow-up of an individual patient". But as I go through the manuscript, it is actually a research article where there is a sample of participants who answered a questionnaire and also some air quality measurements at the workplace. So, why did you choose to submit it as a case report?
2. English language needs editing in many places throughout the manuscript. Especially grammar and use of singular and plural.
3. The citation style is not consistent, as the author uses number on brackets in some places, e.g. [4-7]. And they use superscript in other places, e.g. 8 . Please use the same style throughout the paper.
4. Section 2 in the manuscript is titled “purpose of the study”. I would title it “importance of the study or rationale” instead, since it does not include the study aims or objectives.
5. The objectives are missing in the manuscript. What exactly are your specific objectives?
6. The authors should clearly state the ethical considerations in their methods section. This should include how they made sure to deal with al potential ethical issues, including voluntary participation, confidentiality, risks and benefits and informed consent.
7. Selection of participants: Why you wanted to include only Afro-Caribbean hairdressers? What about bias generated from participants who filled the online survey and claimed they are of this origin but they are actually not? Have you considered this possibility?
8. The indoor air sampling: did you select the sampled salons where the participants answered the survey? Or other salons? And are the sampled salons also operated by Afro-Caribbean hairdressers?
Author Response
Dear Reviewer 1:
Many thanks for the input made to our submission. We have improved the work based on comment raised and provided response to each comment in the table below.
Kind regards
Haruna

Reviewer 2 Report
This paper involves interesting facts on the status of occupational safety and hygiene perception among Afro-Caribbean hair salon operators. Authors also attempted to link the heath disorders of the operators and indoor air quality. However, I think the description on the indoor air measurement is not enough. In the conclusion authors described “Findings from this study revealed that high amount of volatiles are released in the indoor work environment which correspond with high report incidence of occupational ailments…”. Where can we find the findings in this manuscript? So, please address following points before publication.
As for TVOC concentrations shown in Table 4, are the values significantly greater than outdoor of salons, other workplace and/or living environments? Is the TVOC concentration actually greater during the working time? Please show the differences between working time and closed time. According to Oikawa et al. (2012), thioglycolic acid (TGA) is a volatile compound but easily adsorbs on hair, cloth and other surfaces, and hence TGA in indoor air is not important as a possible exposure route for hairdressers and customers in this beauty salon. Please mention such kind of chemicals as a potential hazard in a salon.
Oikawa et al., J Occup Health. 2012;54(5):370-5. DOI: 10.1539/joh.12-0084-fs
Author Response
Dear Reviewer 2,
Many thanks for the input made to our submission. We have improved the work based on comment raised and provided response to each comment in the table below.
Kind regards
Haruna

Round 2
Reviewer 1 Report
No further comments